# Accelerometer-Measured Physical Behavior and Cardiorespiratory Fitness as Indicators of Work Ability

**DOI:** 10.3390/ijerph20075414

**Published:** 2023-04-05

**Authors:** Pauliina Husu, Kari Tokola, Henri Vähä-Ypyä, Harri Sievänen, Tommi Vasankari

**Affiliations:** 1The UKK Institute for Health Promotion Research, Kaupinpuistonkatu 1, FI-33500 Tampere, Finland; 2Faculty of Medicine and Health Technology, Military Medicine, Tampere University, Kalevantie 4, FI-33014 Tampere, Finland

**Keywords:** physical activity, sedentary behavior, time in bed, accelerometer, 24/7 consecutive days, fitness, work

## Abstract

Work ability (WA) reflects an individual’s resources, work demands, and related environment. Self-reports have shown that higher physical activity (PA) is associated with better WA. This study investigated associations of accelerometer-measured (24/7) physical behavior and cardiorespiratory fitness (CRF) with WA. In the FinFit2017-population-based study, the physical behavior of 20–69-year-old working Finns was measured in terms of PA, standing, and sedentariness using validated MAD-APE algorithms based on raw triaxial accelerometer data. During waking hours, the accelerometer was hip-worn, while during the time in bed (TIB), it was worn on the non-dominant wrist. CRF was measured with a 6 min walk test. WA was assessed by four questions excerpted from the Work Ability Index (WAI), called the short WAI (sWAI). Participants (*n* = 1668, mean age 46.6, SD = 10.9, 57% women) scored on average 23.3 on the sWAI (range 6–27), with a higher value indicating a better WA. More minutes in standing (*p* = 0.001) and in moderate (*p* = 0.004) and vigorous PA (*p* < 0.001) as well as a higher step number (*p* < 0.001) and better CRF (*p* < 0.001) were associated with a higher sWAI value. More time spent lying down (*p* < 0.001) and in high-movement (*p* < 0.001) and total TIB (*p* = 0.001) was associated with a lower sWAI. Detailed analysis of 24/7 physical behavior can be utilized in identifying individual-related indicators of WA.

## 1. Introduction

Perceived work ability (WA) reflects a combination of an individual’s capabilities and resources, the demands of the work, and the related environment [1]. WA is a context- and time-bound multidimensional concept developed in the early 1980s [1]. Poor current WA predicts decreased future WA, increased risk for long-term sickness absence [2], thoughts of early retirement from the labor market [3], and early retirement [2]. WA is associated with individual characteristics, lifestyle, demands at work, and physical condition: for example, poor WA has been reported to be associated with older age, obesity, lack of vigorous leisure-time physical activity (PA), poor musculoskeletal capacity, high mental work demands, lack of autonomy, poor physical work environment, and high physical workload [4]. Altogether, WA is modulated by several factors, some of which can be individually affected and modified.

Health-related behaviors are examples of these modifiable factors. Studying the association between PA, smoking, diet, and WA, low PA has been found to have the greatest population attributable fraction for poor WA [5]. Further, WA can be improved by exercise, especially among the individuals with a poor-to-moderate baseline WA [6]. The association between PA and WA seems to be stronger in physically demanding jobs than in sedentary jobs [7]. There seems to be a dose response between moderate-to-vigorous-intensity PA (MVPA) and better WA, but the evidence regarding light PA is inconclusive [8,9]. Most of these previous studies have used self-reported data on PA. Regarding the device-based measurements of PA, Nawrocka et al. [10] reported that adherence to the PA guidelines is related to better perceived WA among middle-aged women. Further, a metabolic equivalent (3.5 mL/kg/min of oxygen consumption, MET)-based intensity of PA seems to be positively associated with WA [11], with a higher PA intensity being associated with better WA.

Although several studies have reported associations between PA and WA, much less is known about the association between sedentary behavior (SB) and WA. SB is known to be harmful to health and wellbeing [12]. Giurgiu et al. [11] reported that a greater number of sit-to-stand transitions was positively associated with WA, whereas lack of sleep deviating from 8 h and SB bouts exceeding 20 min had negative associations with WA. Reducing SB during working hours has been found to have a beneficial effect on perceived health and neck and shoulder pain as well as on perceived vitality in work engagement and work performance [13].

In addition to PA, physical fitness may also increase an individual’s capacities to cope with the demands of everyday life, including work [14]. A recent study showed that cardiorespiratory fitness (CRF) has declined in most occupational groups over the last two decades, being the greatest in blue-collar and low-skilled occupational groups [15]. CRF, especially when assessed with walking-based methods, correlates with WA [16,17]. However, high fitness may not automatically improve WA [18]. Walking is a basic function needed in many daily activities [17], and it is also the most popular type of PA in Finland where this study was conducted [19]. Thus, it would be important to assess CRF with walking-based methods when studying WA.

Previous studies using accelerometer measurements to assess the association between 24/7 physical behavior and WA are scarce [10,11], and even fewer studies have also employed a walking-based test to assess CRF and its associations with WA. Thus, the present study aimed at investigating how a short work ability index (sWAI) [20] and a single-item work ability score (WAS) describing the current WA in relation to the lifetime best are associated with 24/7 physical behavior and measured CRF among a population-based sample of working adults. The hypothesis was that less PA, regardless of intensity, more SB, and poorer CRF would be associated with poorer WA.

## 2. Materials and Methods

This study is based on the population-based FinFit2017 study, which is a cross-sectional, multifactorial study on PA, fitness, and health conducted with a stratified random sample of 20–69-year-old Finnish adults. Potential participants were drawn from the population registry in seven city-centered regions of Finland: 300 men and 300 women from both Helsinki and Tampere regions and 150 men and 150 women from each of the Turku, Kuopio, Jyväskylä, Oulu, and Rovaniemi regions spread across five age groups (20–29, 30–39, 40–49, 50–59, and 60–69 years). Other inclusion or exclusion criteria were not used. Invitation letters containing information about the study and informed consent with the option to withdraw from the study at any time were mailed to all 13,500 potential participants belonging to the sample. The data collection was conducted between September 2017 and March 2019, and the study comprised three parts: (1) a questionnaire assessing health status, including depressive symptoms; (2) a health examination, including blood samples, anthropometric measurements, and a 6 min walk test [21]; and (3) 24/7 measurement of physical behaviors with a triaxial accelerometer [22]. Descriptive results of the accelerometer measurements have been reported previously [22].

Inclusion criteria for the present study were 20–69-years of age, living in one of the above-mentioned city-centered regions, being at full- or part-time work or an entrepreneur, and responses to sWAI and WAS questions. The present study included 1668 participants (57% women, mean age of 46.6 years, standard deviation (SD) of 10.9). The study was carried out in accordance with the Declaration of Helsinki. The Regional Ethics Committee of the Expert Responsibility Area of Tampere University Hospital approved the study (R17030). All participants gave signed informed consent before participation.

### 2.1. Work Ability

WA was assessed by four questions excerpted from the seven-item Work Ability Index (WAI) [23]. The WAI and its items provide a valid instrument to assess WA in clinical occupational health and research [23]. Both the whole seven-item WAI and its individual items predict work disability, retirement, and mortality [24]. The reliability of WAI has been shown to be acceptable [25] Four items included in the sWAI were the following: (1) current work ability in comparison to the lifetime best on a visual analog scale (VAS, 0–10), where 0 = unable to work and 10 = the best possible; (2) work ability in relation to physical work demands (score of 1–5, 1 = very poor, 5 = very good); (3) work ability in relation to mental work demands (score of 1–5, 1 = very poor, 5 = very good); and (4) personal prognosis for work ability in 2 years’ time (1 = hardly able to work, 4 = not sure, 7 = almost certain work ability). A summary score of these 4 items (range of 3–27) was calculated. In addition to the summary of the sWAI, we also used a single-item question of current WA (work ability score (WAS), range of 1–10) as an indicator of present WA. Previously, the same version of the sWAI has been used among health care workers with recurrent low back pain [20].

### 2.2. Physical Behavior

Participants’ physical behavior in terms of time in bed (TIB), SB (lying, reclining, sitting), standing, light PA, moderate PA, and vigorous PA was measured by a tri-axial accelerometer (UKK RM42, UKK Terveyspalvelut Oy, Tampere, Finland) 24/7. During waking hours, the accelerometer was attached to an elastic belt and worn on the right side of hip, excluding water-based activities. For the assessment of TIB, the accelerometer was moved from the belt to an adjustable wristband and attached to the non-dominant wrist on the knuckles’ side. The accelerometer collected and stored the raw triaxial data in actual g-units in ±16 G range at a 100 Hz sampling rate [22].

During waking hours, the mean amplitude deviation (MAD) was calculated from the resultant acceleration signal in six-second epochs, which is a valid indicator of incident oxygen consumption during locomotion [26]. The epoch-wise MAD values were converted to METs, and intensity was calculated as the one-minute exponential moving average of epoch-wise MET values. Light PA was defined as MET values higher than or equal to 1.5 but less than 3.0 (MAD value between 22.5 and 91.5 mg) and MVPA as MET values higher than or equal to 3.0 (MAD over 91.5 mg) [27]. The body posture (lying, reclining, sitting, standing) was determined for epochs during which MAD values were lower than 22.5 mg [28].

The epoch-wise values representing lying, reclining, sitting, or standing periods were also smoothed by a one-minute exponential moving average. The determination of the body posture was based on the angle for posture estimation (APE) method where incident accelerometer orientation was assessed in relation to the reference gravity vector [28]. The posture was classified as standing if the APE was less than 11.6°, sitting if the APE was between 11.6° and 30.0°, reclining if the APE was between 30.0° and 73.0°, and lying otherwise.

The analysis of TIB was based on the movement of the non-dominant wrist when the accelerometer was wrist-worn. The bed-in time (time going to bed for sleeping) was defined as the moment when the accelerometer was moved from the hip to the wrist and the bed-out time (time to wake up) when it was moved from the wrist to the hip. The sensor location was determined from the amount and frequency of changes in the accelerometer orientation. Thus, the TIB is the duration between the bed-out and bed-in times. The analysis method calculates the frequency of the wrist movements and categorizes the TIB into the high, medium, and low-movement categories. The method is described in more detail in Husu et al. [22].

### 2.3. Cardiorespiratory Fitness

Cardiorespiratory fitness (CRF) was assessed at the health examination by the 6 min walk test [29]. The health examinations were conducted by the regional research teams consisting of 4–5 educated research assistants. The research assistants asked the participants to walk back and forth along the 15 m walking track as fast as possible for 6 min and recorded their heart rate with a heart rate monitor (Polar M61, Polar Electro, Kempele, Finland). Weight in kilograms and height in meters were measured and used as an indicator of body composition, the body mass index (BMI, kg/m^2^). The measurements were conducted with light clothing without shoes before the 6 min walk test. For men, the estimation of maximal oxygen consumption (VO_2_peak) was based on the distance walked in 6 min, age, BMI, heart rate, and height; for women, it was based on the distance walked in 6 min, weight, and age [21].

### 2.4. Background Characteristics

Data on working status, education, and the physical demands of the work were collected by the same self-reported questionnaire as the sWAI and WAS. Working status was assessed by asking the participants to report whether they were a part- or full-time employee, entrepreneur, retired, unemployed, on parental leave, or a student at the time of completing the questionnaire. Only the respondents who reported to be a part- or full-time employee or entrepreneur (*n* = 1668) were included in the present analysis. Education was assessed by asking the participants to report their highest education level they had completed after the primary school. The responses were categorized into four groups: (1) no vocational education, (2) vocational education, (3) Bachelor’s degree, (4) Master’s degree or higher. The physical demands of the work were assessed by asking the participants to report how physically demanding the work they had been doing during the past 12 months was. The response options were (1) light sedentary work, (2) other sedentary work, (3) light standing work, (4) light to moderate physically active work, and (4) vigorous or very vigorous physically active work.

### 2.5. Statistical Analysis 

The associations between accelerometer-measured physical behavior and sWAI were analyzed by Pearson partial correlations controlled for age and sex and general linear models, more specifically by gamma regression, adjusted first for age group, sex, education, and the physical demands of the work. Further, the analyses were conducted by adjusting for tertiles of VO_2_peak indicating CRF and BMI. The association between VO_2_peak and sWAI was analyzed by gamma regression adjusted for age group, sex, education, and the physical demands of the work. The corresponding analyses were conducted by using the single-item WAS instead of the sWAI as an indicator of WA. Results from gamma regression are presented as coefficient B and exponential of coefficient B with 95% Wald confidence intervals All analyses were performed by IBM SPSS Statistics for Windows, Version 28.0.

## 3. Results

The participants (*n* = 1668) scored on average 23.3 (range 6–27, SD 2.9) points in the sWAI, with a higher value indicating better WA. Most of the participants used the accelerometer for at least 4 days, for 24 h each day (*n* = 1255). Half (49.9%) of the participants had sedentary work and 22.7% reported having light standing work. In contrast, 14.8% had light to moderate physically active work and only 12.6% of the participants had vigorous physically active work. Over one-fourth (27.1%) of the participants had a Master’s degree, 22.4% had a Bachelor’s degree, 44.7% reported vocational education as their highest educational level, and nearly 5.8% had only elementary school education.

The participants (*n* = 1255) were sedentary on average 9 h of the day, 4 h 56 min of which were spent reclining, 2 h 52 min sitting, and 1 h 13 min lying. Two hours and 2 min were spent standing, 3 h 51 min in light PA, 45 min in moderate PA, and only 3 min in vigorous PA. Participants took on average 7845 steps per day. Total TIB covered on average 8 h 16 min, 2 h 34 min of which was categorized as low movement, 4 h 17 min as medium movement, and 1 h 25 min as high movement. Of the 1668 participants, 1137 (57% women) completed the 6 min walk test. The mean VO_2_peak estimated from the test was 34.5 mL/kg/min, 37.2 mL/kg/min among the men and 32.4 mL/kg/min among the women. (Table 1).

Of the accelerometer variables, sitting (partial correlation = 0.09, *p* = 0.001), standing (partial correlation = 0.11, *p* < 0.001), moderate PA (partial correlation = 0.08, *p* = 0.006), vigorous PA (partial correlation = 0.15, *p* < 0.001), and daily steps (partial correlation = 0.09, *p* = 0.001) had positive correlations with the sWAI indicating that longer time in these behaviors or higher step number were associated with a higher sWAI value and thus better WA. Lying down during waking hours (partial correlation = −0.13, *p* < 0.001) and total TIB (partial correlation = −0.12, *p* < 0.001) showed negative partial correlations indicating that longer time in these behaviors was associated with poorer WA.

Gamma regression analysis adjusted for age group, sex, education, and the physical demands of the work confirmed these findings: longer time spent standing and in moderate and vigorous PA as well as higher daily step number were associated with a higher sWAI value. Longer time spent lying down during waking hours and longer TIB were associated with a lower value in the sWAI (Table 2).

When analyzing the association between CRF and WA, excluding the physical behavior parameters, the VO_2_peak had a positive partial correlation (0.20, *p* < 0.001) with the sWAI indicating that higher fitness is associated with better WA. Also according to gamma regression adjusted by age group, sex, education, and the demands of the work, better fitness was associated with a higher sWAI value. When VO_2_peak was categorized by the age-group- and sex-specific tertiles, 352 participants belonged to the low-fit group, 375 to the mid-fit group, and 410 to the high-fit group. Both the mid-fit and high-fit groups were more likely to have higher sWAI values, indicating better WA compared to the low-fit group (Table 2).

When corresponding analyses were conducted separately in the five age groups, among the 40–49 and 50–59-year-old participants longer time spent in moderate and vigorous PA and in high-movement TIB, higher daily step number, and better CRF had corresponding statistically significant associations with sWAI, as the main findings presented in Table 2: higher activity and better fitness were associated with better WA, and longer time in high-movement TIB was associated with poorer WA. Among the youngest and the oldest participants, there were no corresponding systematic associations (Table 3).

Since both CRF and BMI may affect WA, we also analyzed the adjusted associations between physical behavior and sWAI, respectively (Table 4). The participant’s BMI was on average 26.2 (SD 4.3) kg/m^2^ and 17% had a BMI exceeding 30 kg/m^2^. When adjusting the gamma regression analysis for CRF tertiles, moderate PA was no longer statistically significantly associated with sWAI, but the other associations remained similar. The associations also remained after adjusting for BMI.

In addition to using the sWAI as the outcome variable, all analyses were conducted with a single-item WAS as an indicator of WA. The results remained very similar with this outcome: a longer time spent standing and in moderate and vigorous PA and a higher number of daily steps were associated with higher WAS indicating better WA, while a longer time spent lying down during waking hours and a longer total TIB were associated with lower WAS. Moreover, the association between CRF and WAS was similar to the one between CRF and sWAI: higher fitness was associated with higher WAS indicating better WA.

## 4. Discussion

The findings of the present study showed that longer times of accelerometer-measured moderate and vigorous PA and standing as well as better CRF were associated with better perceived WA. In contrast, longer time spent lying down during waking hours and longer total TIB were associated with poorer WA. More active individuals and those with better CRF may be able to cope with their daily routines with less physical effort and thus have better perceived WA. More sedentary individuals, in turn, may have poorer abilities to respond to the demands of daily life and perceive poorer WA. 

Although several previous studies have shown the association between low PA and poor WA [5,7,8,9,10,11], there seems to be a lack of population-based studies analyzing how the accelerometer-measured physical behavior and CRF are associated with WA. Most of the previous studies have used self-reported data on PA, but SB, standing, and TIB have rarely been included. The present study provides new information to be utilized in future studies as well as in the practice of health promotion among the working-age population. According to the present findings, a detailed analysis of 24/7 physical behavior can be utilized in identifying individual-related indicators of WA.

In contrast to a significant association between MVPA and WA, light PA was not associated with WA in the present cross-sectional analysis, which is in line with findings by Calatayud et al. [9]. Using the device-based data, Nawrocka et al. also [10] reported a positive association between MVPA and WA. In a prospective study based on self-reported PA data by Arvidson et al. [8], light PA was associated with improved WA, but moderate and vigorous PA were more strongly related to a positive change in WA [8].

When analyzing the association between physical behavior and WA, both education and physical demands of the work were significant confounders. Education had a linear association with WA indicating that higher education was associated with better WA. Physical demands of the work were non-linearly associated with WA, when only the most physically demanding work showed a statistically significant association with poorer WA. Regarding the analysis between CRF and WA, the physical demands of the work factor was a statistically significant confounder, but education was not. 

When the associations were evaluated separately in the five age groups, the findings regarding moderate and vigorous PA, step number, and CRF among the 40–59-year-old participants were statistically significant and congruent with the main results (Table 2). Among the youngest and the oldest participants, no systematic associations were found. The number of participants was highly reduced in the sub-analyses, which may have affected the levels of statistical significance. Further, the occupational status of the youngest and the oldest participants differed from that of the middle-aged participants, with part-time work being more common among the extreme groups. Older participants are also more likely to have a poor health status and declined functional capacity than the younger ones [30], which may affect their perception of WA. 

Higher time in both high-movement and total TIB were associated with poorer WA, while medium- and low-movement TIB were not. Very short [31] or very long total TIB seems to be associated with poorer WA, but this was not verified in the present analysis. Most of the participants of the present study had a total TIB of around 8 h per day. There were very few participants with very short or very long total TIB, so we were not able to analyze the associations of these extremities with WA. Further, we were only able to consider the wrist movement during the total TIB, not the actual sleep quantity or quality. 

We used a short work ability index (sWAI) as the main indicator of WA [20]. The index includes four questions from the original seven-item Work Ability Index (WAI) [23]. Since the sWAI has not been used as widely as the original WAI, we also conducted the analysis by using a single question of current WA score (WAS) as the outcome. According to present findings, the associations between both physical behavior and WAS and CRF and WAS were very similar to the ones for sWAI. This is in line with previous findings [32] indicating that WAI and WAS represent a similar construct. Convergent validity between the single-item WAS and WAI has been reported [32], and WAS has shown equal validity against disability pensions as WAI [33]. Ebener and Hasselhorn [34] analyzed WAS and two items assessing WA in relation to the mental and physical demands of the job against each other and the WAI. They found that WAS and the two items correlated moderately, and to a similar degree, with constructs such as self-rated general health, burnout, and consideration of leaving the profession. 

The 24/7 measurement of physical behaviors (low, medium, and high movements during TIB and SB, standing, light PA, and MVPA during waking hours) is the main strength of the present study. We used two body sites, depending on the purpose of the measurement, to attach the accelerometer. Further, a population-based sample of seven urban and suburban areas covering a wide age range of working adults from 20 to 69 years, the evaluation of CRF [21], and the analysis of the associations of physical behavior and CRF with two indicators of WA (sWAI and WAS) can be considered other strengths of the study. 

The main weakness of the study is the cross-sectional design, which does not allow any causal interpretation of the findings. Secondly, although we used several covariates in the analyses, it is possible that some other confounding factors, not recognized in this study, may have affected the associations found. Further, only 42% of the sample was reached, and 51% of the reached persons agreed to participate in the study, which indicates selective participation [22]. The study participants were more educated than the general population in Finland [35], which may have affected the present findings [22]. Moreover, the contexts of PA and SB could not be identified, and the total time in bed and wrist movements instead of actual sleep analysis were used to characterize the sleep period.

## 5. Conclusions

It is important to consider the multidimensional nature of WA in maintaining and promoting work performance and wellbeing [4]. The associations identified in the present study can be utilized when planning and conducting these actions. Since several physical behaviors were significantly associated with WA, the changes in these behaviors may have a great potential to enhance WA. In the future, the potential causality of the found associations needs to be confirmed in longitudinal study designs.

## Figures and Tables

**Table 1 ijerph-20-05414-t001:** Participants’ daily physical behavior and cardiorespiratory fitness broken down by age group and sex.

		Age Groups					
		20–29	30–39	40–49	50–59	60–69	Total
Men	*n*	36	110	136	160	67	509
		mean (SD)	mean (SD)	mean (SD)	mean (SD)	mean (SD)	mean (SD)
Physical activity	light (h:min)	3:40 (1:10)	3:55 (1:17)	4:00 (1:16)	3:41 (1:09)	3:27 (0:58)	3:47 (1:12)
	moderate (h:min)	0:50 (0:21)	0:44 (0:19)	0:46 (0:20)	0:49 (0:28)	0:49 (0:25)	0:47 (0:23)
	vigorous (h:min)	0:04 (0:08)	0:04 (0:08)	0:05 (0:09)	0:03 (0:07)	0:02 (0:05)	0:04 (0:08)
	steps (steps)	8020 (2383)	7417 (2217)	8025 (2467)	7902 (3031)	7752 (2942)	7819 (2667)
Standing	(h:min)	1:27 (0:37)	1:47 (0:42)	1:50 (0:46)	1:46 (0:59)	1:50 (0:46)	1:47 (0:49)
Sedentary behavior	lying (h:min)	1:24 (0:51)	1:21 (0:45)	1:14 (0:42)	1:20 (0:52)	1:10 (0:43)	1:18 (0:47)
	reclining (h:min)	5:25 (1:18)	5:04 (1:18)	4:58 (1:17)	5:18 (1:30)	5:47 (1:27)	5:13 (1:24)
	sitting (h:min)	2:48 (1:09)	2:48 (0:54)	2:54 (1:05)	2:50 (1:09)	3:03 (0:59)	2:52 (1:04)
Time in bed	high movement * (h:min)	1:46 (0:45)	1:29 (0:40)	1:30 (0:41)	1:28 (0:35)	1:27 (0:41)	1:30 (0:39)
	medium movement * (h:min)	4:37 (0:44)	4:26 (0:41)	4:27 (0:49)	4:16 (0:51)	4:09 (0:46)	4:22 (0:48)
	low movement * (h:min)	1:58 (0:39)	2:22 (0:40)	2:14 (0:40)	2:29 (0:45)	2:20 (0:45)	2:20 (0:43)
Total time in bed	(h:min)	8:21 (0:54)	8:18 (1:04)	8:11 (1:13)	8:13 (1:09)	7:56 (1:07)	8:12 (1:08)
Cardiorespiratory fitness	*n*	36	106	133	151	63	489
	VO_2_peak (mL/kg/min)	41.2 (6.1)	40.0 (4.4)	38.3 (5.6)	35.0 (4.8)	33.1 (5.2)	37.2 (5.7)
Women	*n*	45	140	212	248	101	746
		mean (SD)	mean (SD)	mean (SD)	mean (SD)	mean (SD)	mean (SD)
Physical activity	light (h:min)	3:56 (1:13)	3:51 (1:12)	3:59 (1:17)	3:55 (1:20)	3:42 (1:25)	3:54 (1:16)
	moderate (h:min)	0:56 (0:25)	0:42 (0:21)	0:45 (0:23)	0:43 (0:23)	0:41 (0:25)	0:44 (0:23)
	vigorous (h:min)	0:04 (0:05)	0:05 (0:08)	0:04 (0:06)	0:02 (0:06)	0:01 (0:04)	0:03 (0:06)
	steps (steps)	8895 (3079)	7843 (2779)	8122 (2981)	7709 (2781)	7264 (2965)	7863 (2898)
Standing	(h:min)	1:56 (0:36)	2:08 (0:57)	2:19 (0:59)	2:13 (0:59)	2:11 (1:02)	2:13 (0:58)
Sedentary behavior	lying (h:min)	1:28(0:36)	1:13 (0:36)	1:09 (0:41)	1:08 (0:39)	1:55 (0:40)	1:10 (0:39)
	reclining (h:min)	4:39 (1:11)	4:39 (1:15)	4:44 (1:27)	4:29 (1:09)	4:46 (1:25)	4:45 (1:24)
	sitting (h:min)	2:34 (0:53)	2:53 (0:57)	2:48 (0:57)	2:52 (1:01)	3:08 (1:18)	2:52 (1:02)
Time in bed	high movement * (h:min)	1:20 (0:30)	1:24 (0:39)	1:15 (0:33)	1:23 (0:35)	1:27 (0:34)	1:21 (0:47)
	medium movement * (h:min)	4:39 (0:42)	4:29 (0:50)	4:16 (0:46)	4:06 (0:42)	4:01 (0:47)	4:15 (0:47)
	low movement * (h:min)	2:28 (0:46)	2:35 (0:43)	2:41 (0:47)	2:48 (0:48)	2:55 (0:47)	2:44 (0:54)
Total time in bed	(h:min)	8:27 (0:49)	8:28 (1:13)	8:13 (1:06)	8:17 (1:04)	8:24 (1:06)	8:20 (1:06)
Cardiorespiratory fitness	*n*	40	117	186	220	85	648
	VO_2_peak (mL/kg/min)	35.2 (5.0)	34.7 (5.6)	33.4 (6.0)	30.9 (5.7)	30.2 (5.6)	32.4 (6.0)

* of the non-dominant wrist.

**Table 2 ijerph-20-05414-t002:** Gamma regression on the association between physical behavior, cardiorespiratory fitness, and work ability (sWAI) adjusted for age group, sex, education, and physical work demands.

Physical Behavior		B	Exp (B)	95% CI	*p* Value
Physical activity	light per 60 min	0.007	1.007	1.00–1.01	0.060
	moderate per 60 min	0.028	1.029	1.01–1.05	0.004
	vigorous per 60 min	0.134	1.144	1.07–1.22	<0.001
	steps per 1000	5.35 × 10^−6^	1.005	1.00–1.01	<0.001
Standing	per 60 min	0.010	1.014	1.00–1.02	0.001
Sedentary behavior	lying per 60 min	−0.020	0.980	0.97–0.99	<0.001
	reclining per 60 min	−0.00	0.998	0.99–1.00	0.508
	sitting per 60 min	0.007	1.007	1.00–1.02	0.058
Time in bed	high movement * per 60 min	−0.025	0.975	0.96–0.99	<0.001
	medium movement * per 60 min	−0.007	0.993	0.98–1.00	0.141
	low movement * per 60 min	−0.010	0.992	0.98–1.00	0.092
	total per 60 min	−0.001	0.990	0.99–1.00	0.001
Cardiorespiratory fitness	VO_2_peak (mL/kg/min) per 10 mL VO_2_peak	0.004	1.004	1.00–1.01	<0.001
	low-fit tertile	ref.			
	mid-fit tertile	0.025	1.025	1.01–1.05	0.013
	high-fit tertile	0.046	1.047	1.03–1.07	<0.001

* of the non-dominant wrist; B = coefficient from gamma regression, Exp(B) = exponential of B, CI = confidence interval.

**Table 3 ijerph-20-05414-t003:** Gamma regression on the age-group-specific association between physical behavior and work ability (sWAI) adjusted for sex, education, and physical work demands.

	Age Group	20–29		30–39		40–49		50–59		60–69	
Physical behavior		B	*p* value	B	*p* value	B	*p* value	B	*p* value	B	*p* value
Physical activity	light per 60 min	0.009	0.496	0.010	0.111	−1.34 × 10^−4^	0.952	0.012	0.078	0.008	0.480
	moderate per 60 min	−0.013	0.683	0.058	0.006	0.037	0.038	0.037	0.020	−0.022	0.484
	vigorous per 60 min	0.062	0.607	0.104	0.054	0.117	0.029	0.220	<0.001	0.158	0.373
	steps per 1000	3.12 × 10^−4^	0.947	9.88 × 10^−3^	<0.001	4.62 × 10^−3^	0.049	8.18 × 10^−3^	0.001	−5.42 × 10^−3^	0.740
Standing	per 60 min	0.024	0.259	4.92 × 10^−3^	0.544	0.008	0.258	0.014	0.050	0.031	0.039
Sedentary behavior	lying per 60 min	−0.016	0.358	0.006	0.556	−0.015	0.120	−0.039	<0.001	−0.014	0.482
	reclining per 60 min	2.82 × 10^−3^	0.795	−5.60 × 10^−3^	0.321	5.62 × 10^−3^	0.240	−5.19 × 10^−3^	0.275	−0.007	0.494
	sitting per 60 min	0.020	0.132	−4.53× 10^−3^	0.553	4.76 × 10^−3^	0.489	0.010	0.116	0.010	0.372
Time in bed	high movement * per 60 min	−0.016	0.465	−0.011	0.277	−0.032	0.003	−0.039	0.001	−0.024	0.271
	medium movement * per 60 min	−0.051	0.003	−2.96 × 10^−3^	0.741	−0.010	0.221	2.29 × 10^−3^	0.795	−0.014	0.423
	low movement * per 60 min	5.15 × 10^−3^	0.783	−0.014	0.142	−5.19 × 10^−3^	0.558	−2.95 × 10^−3^	0.733	−0.019	0.284
	total per 60 min	−0.042	0.004	−0.010	0.085	−0.016	0.005	−0.011	0.071	−0.022	0.063
	*n*	81		248		346		406		168	
Cardiorespiratory fitness	VO_2_peak (ml/kg/min), per 10 mL VO_2_peak	0.004	0.863	0.003	0.869	0.027	0.017	0.073	<0.001	0.048	0.093
	low-fit tertile	ref.									
	mid-fit tertile	0.004	0.884	0.001	0.940	0.036	0.032	0.046	0.009	0.004	0.928
	high-fit tertile	8.196 × 10^−5^	0.998	0.002	0.927	0.039	0.013	0.091	<0.001	0.036	0.345
	*n*	76		221		318		370		148	

* of the non-dominant wrist; B = coefficient from gamma regression.

**Table 4 ijerph-20-05414-t004:** Gamma regression on the association between physical behavior and work ability (sWAI) adjusted for age group, sex, education, physical work demands, and cardiorespiratory fitness thirds (A) and for age group, sex, education, physical work demands, and body mass index (B).

		(A)				(B)			
Physical Behavior		B	Exp (B)	95% CI	*p* Value	B	Exp (B)	95% CI	*p* Value
Physical activity	light per 60 min	0.005	1.005	1.00–1.01	0.243	0.001	1.006	1.00–1.01	0.121
	moderate per 60 min	0.018	1.019	1.00–1.04	0.084	0.022	1.022	1.00–1.04	0.039
	vigorous per 60 min	0.110	1.106	1.03–1.19	0.004	0.006	1.123	1.05–1.20	0.001
	steps per 1000	0.004	1.004	1.00–1.01	0.007	0.005	1.005	1.00–1.01	0.002
Standing	per 60 min	0.010	1.011	1.00–1.02	0.019	0.011	1.011	1.00–1.02	0.015
Sedentary behavior	lying per 60 min	−0.018	0.983	0.97–0.99	0.002	−0.018	0.982	0.97–0.99	0.001
	reclining per 60 min	−0.001	1.001	0.99–1.01	0.671	0.000	1.000	0.99–1.01	0.897
	sitting per 60 min	0.004	1.004	1.00–1.01	0.278	0.005	1.005	1.00–1.01	0.249
Time in bed	high movement * per 60 min	−0.014	0.986	0.97–1.00	0.037	−0.016	0.984	0.97–1.00	0.015
	medium movement * per 60 min	−0.007	0.993	0.98–1.00	0.179	−0.001	0.993	0.98–1.00	0.191
	low movement * per 60 min	−0.010	0.990	0.98–1.00	0.052	−0.011	0.989	0.98–1.00	0.039
	total per 60 min	−0.013	0.988	0.98–1.00	0.001	−0.013	0.987	0.98–0.99	<0.001

* of the non-dominant wrist; B = coefficient from gamma regression, Exp(B) = exponential of B, CI = confidence interval.

## Data Availability

The data are maintained at the UKK Institute. The datasets analyzed in the present study are not publicly available due to ethical restrictions (the Regional Ethics Committee of the Expert Responsibility Area of Tampere University Hospital), but more detailed information on the data is available from the corresponding author on reasonable request.

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
