# Peer review of "Accelerometer-Measured Physical Behavior and Cardiorespiratory Fitness as Indicators of Work Ability"

_ijerph, 2023, doi:10.3390/ijerph20075414_

Round 1
Author Response
Dear Reviewer 1 and Ms Yoochareon
Thank you for considering our manuscript (ijerph-2297997) to be published in the IJERPH. Enclosed please find the revised version of the manuscript, which has been modified along the valuable comments suggested by the reviewers. The comments were carefully checked through and the responses submitted to reviewers will clarify the modifications made. In the revised manuscript file, the changes are marked with MS word track changes. We hope that these revisions improved the quality of our manuscript so that it becomes acceptable for publication in the IJERPH. However, if we have not adequately covered the points raised by the reviewers, we are willing to modify the manuscript further.
Sincerely,
Pauliina Husu, PhD
pauliina.husu@ukkinstituutti.fi
Reviewer 1:
General Comments
The manuscript titled above addresses the important contribution of physical activity to favorable productive environments, reinforcing its functionalist perspective. While the findings aren't particularly groundbreaking, the work itself has potential. I want to make it clear that at no time do I have the right to denigrate the authors' work. My comments have been submitted
to improve work which is generally well written. Specific comments have been added throughout this opinion. My decision was forwarded to the Editor-in-Chief.
Answer: Thank you very much for your general comment. We found your comments very helpful improving our work.
Abstract
Please remove the itemized titles in the abstract (for example Background, Methods..).
Answer: This has been revised as advised.
Introduction
- Establishing concepts - as was done in the first sentence of the introduction - requires theoretical support. Throughout the manuscript, some sentences were presented without the necessary theoretical support. For example: “Perceived work ability (WA) ... and related environment.” Review this point throughout the document and make the necessary adjustments.
Answer: The text has been reviewed throughout and theoretical support/references has been added.
- Measures of sedentary behavior in the context of the study are a strong point that I must highlight. However, it is imperative that the authors present (albeit according to Giurgiu et al.) the bad mechanisms of the relationship between SB and WA. This has been done nicely in relation to physical activity in paragraph 2 (lines 40-51).
Answer: Thank you for pointing this out. New references have been added here to indicate harmful relationship between WA and general health (Biswas et al.) as well as the beneficial effect of reducing SB during working hours on perceived health and work efficiency (Ma et al.).
- A justification based on the WA and not on the use of good instruments needs to be presented.
Answer: We renamed the WA instrument used as a short WAI (sWAI), since it is a shortened version of the original 7-item WAI. The same version of the sWAI has been used among health care workers with recurrent low back pain by Virkkunen et al. This reference has been added to the manuscript.
- Why might the findings of the proposed investigation improve WA?
Answer: This is a cross-sectional study aimed at presenting associations of accelerometer-measured (24/7) physical behavior and cardiorespiratory fitness (CRF) with WA. Unfortunately, we cannot say anything about the causality and thus, not about improving WA either.
- Please present the conceptual hypothesis of the study
Answer: We hypothesize that less PA, regardless of intensity, more SB and poorer CRF would be associated with poorer WA. This has been added to the end of introduction (lines 77-78).
Methods
- Finland is a relatively small country with an estimated population of fewer than six million inhabitants (impressive data). Please, what is the occupationally active Finnish population? If possible, provide the information.
Answer: According to national statistics the number of working individuals was 2 566 000 in 2019. However, we decided not to include this to the manuscript, because it would have needed much more detailed description of the population structure in Finland. The study was conducted with a population-sample based on certain age groups regardless of working status.
- What parameters can be presented to determine the sample size?
Answer: Thank you for this comment. We are very sorry that part of the text was missing from the original manuscript. Now this has been specified in the methods section (lines 82-89) as follows: “Potential participants were drawn from the population registry in seven city-centered regions of Finland: 300 men and 300 women from both Helsinki and Tampere regions and 150 men and 150 women from each of Turku, Kuopio, Jyväskylä, Oulu, and Rovaniemi regions spread across five age groups (20–29, 30–39, 40–49, 50–59, and 60–69 years). Other inclusion or exclusion criteria were not used for the sample. Invitation letters containing information about the study and informed consent with the option to withdraw from the study at any time were mailed to all 13,500 potential participants belonging to the sample.” We hope that this clarifies the sample selection.
- Please make it clear whether or not the 2017 and 2019 participants are different. Due to the limited amount of recruitment information, this point needs to be highlighted.
Answer: The data collection lasted continuously from September 2017 to March 2019, we did not compare the participants between these years.
- What ineligibility criteria were used in recruitment? If possible, show them.
Answer: Inclusion criteria (20-69-years of age, living in one of the selected city-centered regions, being at full- or part-time work or entrepreneur and responses to sWAI and WAS questions) have been added to the methods section (lines 95-97).
- Are there psychometric characteristics of the Work Ability Index (WAI)? If possible, present them objectively.
Answer: The participants scored on average 23.3 (range 6-27, SD 2.9) in the sWAI. This has been presented in the results section (line 194).
Psychometric characteristics of the original Work Ability Index (WAI) have been shown acceptable (Zwart et al. 2022). Despite some differences in individual responses between repeated assessments, there were no significant differences in the mean WAI score at the group level between test and retest measurements (40.4 versus 39.9). The reference with the statement of acceptable reliability has been added to the manuscript (line 107-108). Since the above text concerns the original WAI instead of the sWAI, we decided not to present it in more detail in the manuscript.
Results
- Properly presented. However, it is particularly embarrassing to unite people with a wide age range: 20 to 69 years old, for example. Although the age group was used in the adjusted models, there are several mechanisms that bias the analyses, eg physiological and subjective. Although it is not possible to develop analyses in separate tables for age groups, I suggest pertinent comments on this aspect in the limitations of the study.
Answer: Thank you for bringing up this relevant point. We conducted the analyses by age groups and added a new table 3 to the manuscript. The results showed that among 40-49 and 50-59-year-old participants longer time spent in moderate and vigorous PA and in high movement TIB, higher daily step number and better CRF had corresponding positive, statistically significant associations with sWAI as the main findings presented in the table 2: Higher activity and better fitness were associated with better WA, while longer TIB was associated with poorer WA. Among the youngest and the oldest participants there were no corresponding systematic associations. This has been added to the results section, lines 244-250 and to the discussion section, lines 326-335.
Discussion
-The first paragraph adequately summarizes the main findings of the study. However, is it necessary to present the practical application of this finding? In practice, what do the findings say?
Answer: A clarification has been added to the end of this paragraph (lines 297-300): “More active individuals and those with better CRF may be able to cope with their daily routines with less physical effort and thus have better perceived WA. More sedentary individuals, in turn, may have poorer abilities to respond to demands of daily life and perceive poorer WA.” Practical meaning is presented also at the end of the next paragraph (lines 307-310).
- I recommend deleting the second paragraph, as it does not add to the discussion.
Answer: We would not like to delete the whole paragraph since we believe it offers a rationale for this study. Instead, we deleted one sentence from the paragraph (lines 305-307).
- The main findings of the study were not adequately explained. How do PA, Sedentary behavior, and Cardiorespiratory Fitness explain WA? What is the rationale behind the associations found?
Answer: The rationale behind the findings has been added to lines 297-300: “More active individuals and those with better CRF may be able to cope with their daily routines with less physical effort and thus have better perceived WA. More sedentary individuals, in turn, may have poorer abilities to respond to demands of daily life and perceive poorer WA.”.
- The pooled sample including people aged 20 to 69 be mentioned as a limitation of the study.
Answer: Based on your previous comment and the comments by the reviewer 2, the analyses were now conducted also by age groups and the results have been presented in a new table 3 and on lines 244-250 and discussed on lines 326-335. Thus, this was not mentioned as a limitation of the study.
- The authors used numerous covariates in the adjusted analyses, which does not mean that some residuals did not remain in the final analyses. This information needs to be placed in the constraints.
Answer: Quite agree. We used several covariates that may influence the association between sWAI and physical behavior. However, it is possible that some other confounding factors, not recognized in this study, may have influenced these associations. This has been mentioned in the limitations of the study (lines 366-368).
Reviewer 2 Report
Dear authors,
The topic of this research paper is current, although the actual data collection took place between September 2017 and March 2019. People's physical behavior (physical, mental, social) has a great influence, both in terms of their work capacity and the quality of the work they do. This paper investigates the influence of the physical factor on work capacity. I believe that the paper is very well documented and correctly compiled, and the references are up to date. I believe the article is suitable for publication after making the following changes:
1. Abstract: Please see the journal's guidelines for article preparation (IJERPH Microsoft Word template file). These specify that in the abstract you must remove words: Background, Methods, Results and Conclusions.
2. Keywords: I think it is more appropriate 24h/7 consecutive days (to better understand)
3. Line 49 – MET – please explain what this abbreviation means.
4. In the materials and methods chapter, I did not understand exactly how the number of participants was reported. From the Helsinki and Tampere regions – 300 x 2=600. From the other 5 regions, 150 each, i.e. 150x5=750. I understood correctly? If so, then 600+750=1350 and you stated that 1668 attended. Where did the difference of 318 come from?
Also in this chapter, what were the inclusion and exclusion criteria in this study. Please specify them clearly!
Also here, I think you should specify who conducted this study? From which institution? How many specialists were involved? Was each region managed by a specific research team? etc.
5. Line 139 – „The educated research assistants...” here you are talking about a part of the research team but I think you should specify their number.
6. Line 156 – „There were 11 response alternatives from no vocational education to doctoral degree.” I think/consider this sentence is redundant and should be deleted, especially since it follows the description in the lines below.
7. Lines 176 – 181 – it's not clear enough about the percentages you present. A) 50% + 23% + 13%=86%. What about the remaining 14%? Which category does it fall into? B) 50% + 45% + 6% = 101%. Please clarify this aspect!
8. The interpretation of the data in table 1 is very simplistic. I recommend that they be explained according to the criteria that the authors have established, namely: age group, gender, level of education. Also, I don't understand what is meant by time in bad: high movement, medium movement, low movement? Can you explain to me, please?
9. Lines 191-198 – where does the data presented here come from? I believe that they should be presented broken down by the targeted criteria, to see where these correlations result from.
10. Please explain the data in table 2 according to the indicators presented in its title.
11. Table 3 – I don't understand where the data in the table comes from. Please explain how you achieved this
12. Line 221 – why didn't you present the VO2 peak in the table? A connection must be made between everything you say that there are correlations.
13. In the discussion chapter, nothing appears regarding the results obtained on the age and gender indicators.
Overall recommendations:
- - I believe/ consider that the data should have been analyzed on the presented indicators, respectively on age, sex and education and presented as distinct results.
- Also, to increase the importance of the study, I think a comparative analysis could be done on the extremes of the age range.
Kindest regards!
Author Response
Dear Reviewer 2 and Ms Yoochareon
Thank you for considering our manuscript (ijerph-2297997) to be published in the IJERPH. Enclosed please find the revised version of the manuscript, which has been modified along the valuable comments suggested by the reviewers. The comments were carefully checked through and the responses submitted to reviewers will clarify the modifications made. In the revised manuscript file, the changes are marked with MS word track changes. We hope that these revisions improved the quality of our manuscript so that it becomes acceptable for publication in the IJERPH. However, if we have not adequately covered the points raised by the reviewers, we are willing to modify the manuscript further.
Sincerely,
Pauliina Husu, PhD
pauliina.husu@ukkinstituutti.fi
Reviewer 2:
The topic of this research paper is current, although the actual data collection took place between September 2017 and March 2019. People's physical behavior (physical, mental, social) has a great influence, both in terms of their work capacity and the quality of the work they do. This paper investigates the influence of the physical factor on work capacity. I believe that the paper is very well documented and correctly compiled, and the references are up to date. I believe the article is suitable for publication after making the following changes:
Answer: Thank you very much, your general comment is highly appreciated.
- Abstract: Please see the journal's guidelines for article preparation (IJERPH Microsoft Word template file). These specify that in the abstract you must remove words: Background, Methods, Results and Conclusions.
Answer: This has been revised as advised.
- Keywords: I think it is more appropriate 24h/7 consecutive days (to better understand)
Answer: This has been revised as suggested.
- Line 49 – MET – please explain what this abbreviation means.
Answer: This has been explained. Thank you for pointing this out.
- In the materials and methods chapter, I did not understand exactly how the number of participants was reported. From the Helsinki and Tampere regions – 300 x 2=600. From the other 5 regions, 150 each, i.e. 150x5=750. I understood correctly? If so, then 600+750=1350 and you stated that 1668 attended. Where did the difference of 318 come from?
Answer: Thank you for pinpointing this. We apologize that the last part of the sentence was missing from the original manuscript. The number of invited participants was 300 men and 300 women in each 5 age groups from Helsinki and Tampere and 150 men and 150 women in each 5 age groups from the other 5 regions, adding up to total of 13 500 invited individuals. The inclusion criteria of the present study were 20-69-years of age, living in one of the seven selected city-centered regions, being at full- or pat-time work or entrepreneur and responses to sWAI and WAS questions. Thus, the present study included 1668 participants.
This has been clarified to the text (lines 82-89 and 95-97).
- Also in this chapter, what were the inclusion and exclusion criteria in this study. Please specify them clearly!
Answer: These have been specified (lines 95-97): The inclusion criteria of the present study were 20-69-years of age, living in one of the city-centered regions, being at full- or part-time work or entrepreneur and responses to sWAI and WAS.
- Also here, I think you should specify who conducted this study? From which institution? How many specialists were involved? Was each region managed by a specific research team? etc.
Answer: The health examinations including the 6 min walk test were conducted by the regional research teams consisting of 4-5 educated research assistants. This has been added on lines 155-156. The regional research centers (the LIKES Research Centre for Physical Activity and Health, Jyväskylä; the Department of Sports and Exercise Medicine, Oulu Deaconess Institute Foundation sr, Oulu; the Kuopio Research Institute of Exercise Medicine, Kuopio; the Paavo Nurmi Centre Turku; Santasport, Rovaniemi; city of Helsinki; and Turku University of Applied Sciences) have been acknowledged in the Acknowledgment part of the manuscript (lines 397-401).
- Line 139 – „The educated research assistants...” here you are talking about a part of the research team but I think you should specify their number.
Answer: This has been specified on line 156.
- Line 156 – „There were 11 response alternatives from no vocational education to doctoral degree.” I think/consider this sentence is redundant and should be deleted, especially since it follows the description in the lines below.
Answer: This has been deleted as suggested and the following sentence has been slightly modified: “The responses were categorized into four groups 1) no vocational education, 2) vocational education, 3) Bachelor’s degree, 4) Master’s degree or higher.”
- Lines 176 – 181 – it's not clear enough about the percentages you present. A) 50% + 23% + 13%=86%. What about the remaining 14%? Which category does it fall into? B) 50% + 45% + 6% = 101%. Please clarify this aspect!
Answer: The remaining proportion of participants had light or moderate physically active work. This has been added to the text (lines 197-198). Also, one decimal has been added to the percentages to add them up to 100%, not 101 %, which was due to rounding of numbers.
- The interpretation of the data in table 1 is very simplistic. I recommend that they be explained according to the criteria that the authors have established, namely: age group, gender, level of education. Also, I don't understand what is meant by time in bad: high movement, medium movement, low movement? Can you explain to me, please?
Answer: The table 1 has been divided according to age group and sex. To divide it further according to level of education would enlarge the table 4-fold and the number of participants in some blocks would be only few (for example there were only maximum of 4 women with the lowest education in the three youngest age groups). Therefore, we consider that it is not reasonable to further divide the table 1.
Time in bed (TIB) represents the duration between time going to bed for sleeping (bed-in time) and time to wake up (bed-out time). The bed-in time was defined as the moment when the accelerometer was moved from the hip to the wrist and the bed-out time when it was moved from the wrist to the hip. TIB has been explained in the methods section (lines 144-152): “The analysis of TIB was based on the movement of the non-dominant wrist when the accelerometer was wrist-worn. The bed-in time (time going to bed for sleeping) was defined as the moment when the accelerometer was moved from the hip to the wrist and the bed-out time (time to wake up) was when it was moved from the wrist to the hip. The sensor location was determined from the amount and frequency of changes in the accelerometer orientation. Thus, the TIB is the duration between the bed-out and bed-in times. The analysis method calculates the frequency of the wrist movements and categorizes the TIB into high, medium, and low movement categories. The method is described in more detail by Husu et al. [22].” The clarification about the movement of the non-dominant wrist has been added to the tables 1, 2, and 3 as sub-notes.
- Lines 191-198 – where does the data presented here come from? I believe that they should be presented broken down by the targeted criteria, to see where these correlations result from.
Answer: The data comes from Pearson partial correlations controlled for age and sex. This has been mentioned in the methods section. These numbers are not presented in any table, only in the text. We were not able to break these down by sex, age group and level of education, because it would have enlarged the text 40-fold. However, we included a new table (table 3) which presents the association between sWAI and outcomes of physical behavior in different age-groups.
- Please explain the data in table 2 according to the indicators presented in its title.
Answer: This has been specified in the methods section (lines 189-191) and the parameters have been explained in the tables 2, 3, and 4 as sub-notes.
- Table 3 – I don't understand where the data in the table comes from. Please explain how you achieved this
Answer: Former table 3 is now the table 4. The data comes from the gamma regression analysis. This has been explained in the methods section (lines 189-191) and the means and SDs of CRF have been added to the table 1. The parameters have been explained under the tables 2, 3 and 4.
- Line 221 – why didn't you present the VO2 peak in the table? A connection must be made between everything you say that there are correlations.
Answer: Thank you for asking this. VO2peak has been added to the tables 1, 2 and new table 3.
- In the discussion chapter, nothing appears regarding the results obtained on the age and gender indicators.
Answer: The age group specified analysis were conducted and the results have been added to the new table 3 and on lines 244-250 and discussion on lines 326-335. Sex did not have systematically significant effect on the findings and thus, the analyses were not conducted according to sex groups.
Overall recommendations:
I believe/ consider that the data should have been analyzed on the presented indicators, respectively on age, sex and education and presented as distinct results.
Answer: The analyses were conducted in 5 age groups (20-29, 30-39, 40-49, 50-59 and 60-69) and the results showed that among 40-49 and 50-59-years-old participants longer time spent in moderate and vigorous PA, higher daily step number and better CRF had corresponding positive, statistically significant association with sWAI as the main finding presented in the table 2. Among the youngest and the oldest participants there were not systematically corresponding associations. This has been added to the results section: new table 3 and lines 244-250 and to the discussion section: lines 326-335.
Since sex and education did not have systematically significant effect on the findings, we did not run all the analyses by them. This would have enlarged the results section 5-fold compared to the present one. In addition, some blocks (40 when divided by sex, age groups and education) would have included only very few participants limiting the analysis. The age group and sex -specific numbers of participants have been added to the table 1.
Also, to increase the importance of the study, I think a comparative analysis could be done on the extremes of the age range.
Answer: The associations between physical behavior and WA or CRF and WA were not systematically significant in the youngest and oldest age groups. This has been added to the results (lines 249-250) and discussion (lines 329-335).
Round 2
Reviewer 1 Report
The new version of the manuscript underwent important changes. My opinion has been forwarded to the Editor.
Best regards,
Reviewer 2 Report
I thank you for taking into account my suggestions and I congratulate you for all the effort put into this scientific approach.
Kind regards!